# Investigation on the Performance of Modified Corn Stalk Fiber AC-13 Asphalt Mixture

**Kun Wang** [1], **Lu Qu** [1], **Liang Tang** [2], **Peng Hu** [1,*], **Qiong Wu** [1], **Xiaofei Zhang** [2] and **Hao Xu** [1]

1  Civil Engineering Department, Shandong Jiaotong University, Jinan 250300, China; wangkun@sdjtu.edu.cn (K.W.); shixiaoluya2022@163.com (L.Q.); h15166181493@163.com (Q.W.); 21107019@sti.sdjtu.edu.cn (H.X.)
2  Shandong Road and Bridge Group Co., Ltd., Jinan 250014, China; 15953773666@163.com (L.T.); 18954504822@163.com (X.Z.)
*  Correspondence: 204021@sdjtu.edu.cn

**Abstract:** As an agricultural waste, a large amount of corn stalk will cause environmental pollution. In order to realize the resource utilization of waste and meet the strict requirements of modern traffic on pavement strength and durability, it was modified and applied to an AC-13 asphalt mixture to study its influence on the road performance of asphalt mixture and its mechanism. The road performances of modified corn stalk fiber, lignin fiber, and ordinary asphalt mixtures were evaluated via the wheel tracking test, low-temperature bending test, water immersion Marshall test, freeze–thaw splitting test, and fatigue test. Based on the results of three-point bending fatigue test, the viscoelastic parameters and indexes of the fiber asphalt mixture were obtained by fitting the loading specimen and deflection data with the Burgers constitutive model, and the creep strain response was analyzed by applying dynamic load, so as to explore the relationship between the viscoelastic characteristics and creep behavior of modified corn stalk fiber and AC-13 mixture. The long-term high-temperature performance test of the asphalt mixture with the best fiber content was carried out by using the long-term pavement intelligent monitoring equipment independently developed by the group of investigators. According to the findings, the ideal fiber contents for modified corn and lignin in asphalt mixture are 0.2% and 0.3%, respectively. Among them, the modified corn stalk fiber with a 0.2% content has the best effect on road performance, viscoelastic performance, and the asphalt mixture's creep behavior under dynamic load. Compared with the 0.3% lignin fiber asphalt mixture, its dynamic stability, bending stiffness modulus, immersion residual stability, freeze–thaw splitting strength ratio, and loading times at failure increased by 19.9%, 18.28%, 4.19%, 8.6%, and 9.15%, respectively. Compared with ordinary asphalt mixture, it increased by 47.0%, 28.72%, 7.65%, 15%, and 75.81%, respectively. Moreover, when modified corn stalk fiber is added at 0.2%, the viscoelastic delay time of asphalt mixture is the longest, the strain peak value and rut depth are at a minimum, and the viscoelastic properties, creep properties, and long-term high-temperature properties are the best.

**Keywords:** road engineering; asphalt mixture; modified corn stalk fiber; road performance; fatigue performance; long-term performance



## 1. Introduction

The growing global population and accelerated urbanization are placing higher demands on asphalt pavement performance. Enhancing the quality of the service and durability of pavements and promoting the development of green highways have become the foci of current highway engineering research at home and abroad [1–5]. Fiber is a lightweight, high-strength material that can greatly increase the performances and lifespans of asphalt pavements [6–9]. At present, the most common fibers applied in asphalt mixtures are synthetic and natural fibers. Among them, synthetic fibers mainly include glass fibers,

carbon fibers, and polymer fibers. They can significantly extend the service life of an asphalt pavement made of an asphalt mixture and enhance its performance [10–16]. Natural fibers include animal fibers, plant fibers, and inorganic mineral fibers. Natural fibers have several benefits over synthetic ones, including being renewable, inexpensive, energy-efficient, pollution-free, safe, non-toxic, and biodegradable. Given today's vigorous promotion of green sustainable development, natural fiber offers a distinct competitive advantage in its application since it is a renewable resource with possibilities for sustainable development. Among them, plant fiber has a strong competitive advantage because of its renewable characteristics. Chen et al. [17] noted that the content of cellulose in plant fiber is the highest, and cellulose plays adsorption and stabilization roles in asphalt mixtures due to its excellent fiber strength, stiffness, and structural stability. According to Guo et al. [18], adding modified lignin fiber to asphalt may increase its rutting factor and fatigue life. It can also significantly improve the modified lignin fiber asphalt mixture's high-temperature stability and water stability. Jute fiber can be used to warm mix an asphalt mixture (WMA) to increase its low-temperature fracture resistance, according to Ahmad Mansourian et al. [19]. Shanbara's [20] study found that the cold mix asphalt mixture's (CAM) mechanical characteristics were improved due to the high stiffness and tensile strength of natural coconut shell fiber, thus improving the rutting resistance and an asphalt pavement's resilience cracks. Costa et al. [21] discovered that banana fiber can replace Splitt Mastix Asphalt (SMA) road fiber in asphalt mixture, and it works well at 20 mm in length in terms of enhancing the mechanical qualities of the asphalt mixture. Jia et al.'s study [22] examined how bagasse fiber affected the asphalt mixture's mechanical properties. The findings indicated that the addition of bagasse fiber improved the high-temperature and fatigue properties of the asphalt mixture and the stiffness and crack resistance at medium temperatures. Cheng et al. [23] mixed stalk fiber into asphalt mixture and found that when the fiber content was 4% and the fiber length was 9 mm, the high-temperature and water stability values of the asphalt mixture were the best. In summary, the incorporation of plant fiber can adsorb and stabilize asphalt and enhance the high-temperature and mechanical properties of asphalt mixture. Therefore, it is crucial to research plant fiber, broaden the selection of non-polluting modified performance fiber, promote the development of high-performance asphalt mixtures, and realize the extended lifespans of asphalt pavements.

Corn stalk fiber is a kind of green plant. Due to its inexpensive price, large yield, degradability, and renewability, many scholars have tried to use corn stalk fiber-modified asphalt to improve the asphalt mixture's performance [24,25]. Chen et al. [26] pointed out that compared with polyester fiber, corn stalk fiber leads to a more considerable improvement in the asphalt mortar's performance at low temperatures. Li et al. [27] proposed the best preparation process of corn stalk fiber for road use and found that corn stalk fiber had a stronger impact on improving the asphalt mixture's road performance than lignin fiber. With the passage of time and the deepening of research, many scholars have discovered that the performance of fiber in modified asphalt and asphalt mixture is related to the content and modification effect [28–30]. Wang et al. [31] discovered that the corn stalk fiber modified by NAOH solution improved the adsorption of light components by asphalt, and the asphalt mortar exhibited superior resistance to shear, resistance to deformation at high temperatures, and ability to relax under stress compered to ordinary corn stalk fiber. Chen et al. [32] discovered that the incorporation of modified corn stalk fiber improved the high-temperature deformation resistance and elastic recovery performance of asphalt, and its dispersion uniformity in asphalt mortar was better than that of lignin fiber. Through the comprehensive analysis of the road performance and economy of SMA-13 asphalt mixture, Chen [33] concluded that the ideal content of modified corn stalk fiber is 0.6%. In summary, the research on corn stalk fiber is mainly on the study of ordinary corn stalk fiber-modified asphalt and asphalt mixture. The research on modified corn stalk fiber modified asphalt is mainly from the perspective of asphalt mortar. At the level of asphalt mixture, there are few studies of the performance of AC-13 mixture with modified corn stalk fiber content.

In this paper, the basic characteristics of lignin fibers, ordinary corn stalk fibers, and modified corn stalk fibers were investigated. The optimum asphalt–aggregate ratio of asphalt mixtures with different fiber contents was determined, and the rutting test, low-temperature bending test, immersion Marshall test, freeze–thaw splitting test, and three-point bending fatigue test were carried out to explore the impacts of fiber type and content on the performance of asphalt mixture. The long-term pavement intelligent monitoring equipment independently developed by the research group was used to test the long-term high-temperature performance of the asphalt mixture surface layer with the best fiber content through the scale test to explore the effect of modified corn stalk fiber on the rutting resistance of asphalt pavements.

## 2. Raw Material Performance and Mix Design

### 2.1. Raw Material Properties

#### 2.1.1. Asphalt

The 70# substrate asphalt produced by Sinopec Limited Qilu Branch was used, and its basic properties were tested according to the Chinese Standard JTG E20-2011 [34], as displayed in Table 1.

**Table 1.** Technical specifications for asphalt.

| Performance Indicators | Test Results | Standard Value | Test Methods |
| --- | --- | --- | --- |
| 25 °C penetration (mm) | 69.4 | 60–80 | T0604-2011 |
| Softening point (°C) | 47.09 | ≥46 | T0606-2011 |
| Ductility at 10 °C (cm) | 28.3 | ≥15 | T0605-2011 |
| Ductility at 15 °C (cm) | 160 | ≥100 | T0605-2011 |
| Density (g/cm$^3$) | 1.024 | — | T0603-2011 |
| Flash point (°C) | 275 | ≥260 | T0611-2011 |
| Solubility (%) | 99.8 | ≥99.5 | T0607-2011 |

#### 2.1.2. Aggregates

Coarse aggregate and machine-made sand and mineral powder were all limestone produced in Jinan City, Shandong Province, and their technical indicators met the requirements of the Chinese Technical Specifications (JTG F40-2004) [35].

#### 2.1.3. Fiber

Lignin fiber was produced by Yancheng Denuo Engineering Co., Ltd. (Yancheng, China) Corn stalk fiber and modified corn stalk fiber were prepared in the laboratory through high-speed shear crushing, alkali solution soaking, drying, and other preparation processes [31], as shown in Figure 1. Table 2 displays the technical indications, while Figure 2 displays the appearances and microstructures of the three fibers.

**Table 2.** Fiber technical indicators.

| Fiber Type | Corn Stalk Fiber | Modified Corn Stalk Fiber | Lignin Fiber |
| --- | --- | --- | --- |
| Color | Yellow | Light yellow | Light grey |
| Fiber length/mm | 3.86 | 3.82 | ≤6 |
| PH value | 7.06 | 8.25 | 8.4 |
| Oil absorption multiple/times | 5.71 | 7.41 | 8.13 |
| Heat weight-loss rate (170 °C, 2 h)/% | 7.3 | 6.1 | 5.6 |
| Ash content/% | 5.4 | 5.8 | 18.7 |
| Retention rate/% | 1232 | 1353 | 1368 |

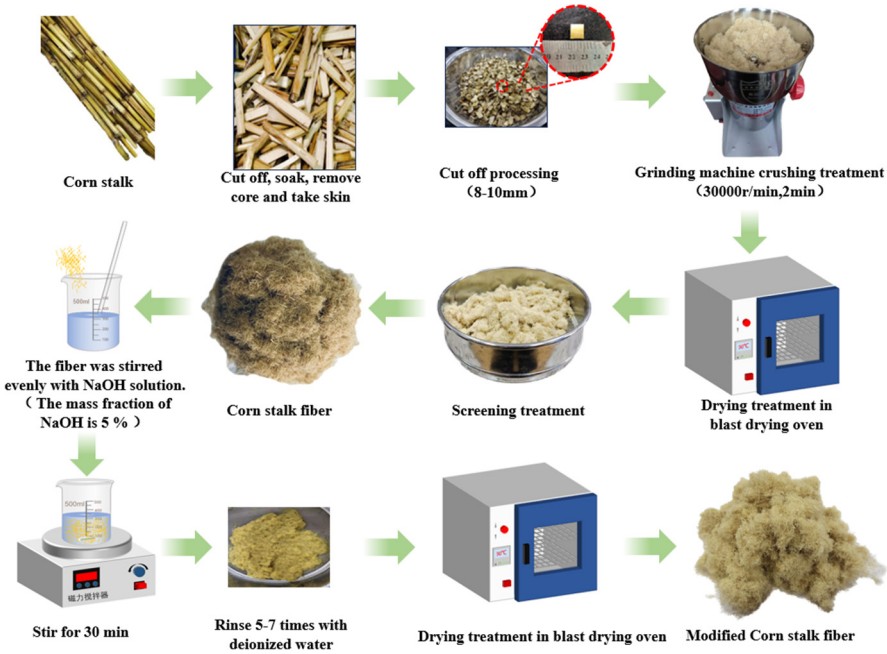

**Figure 1.** Preparation process of corn stalk fiber and modified corn stalk fiber.

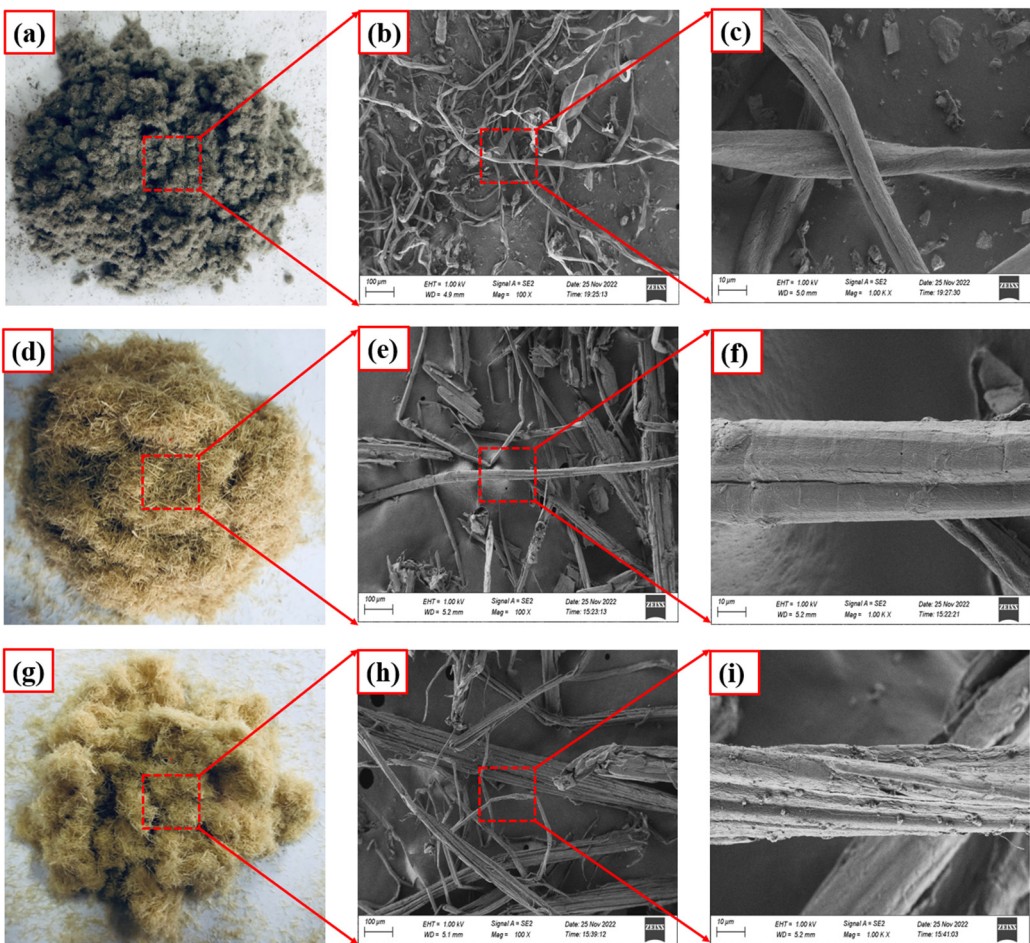

**Figure 2.** Appearance and microscopic morphologies of three different fibers: (**a**) lignin fiber; (**b**,**c**) microscopic morphology of lignin fiber; (**d**) ordinary corn stalk fiber; (**e**,**f**) microscopic morphology of ordinary corn stalk fiber; (**g**) modified corn stalk fiber; (**h**,**i**) microscopic morphology of modified corn stalk fiber.

Figure 2 illustrates that the lignin fiber was mainly composed of monofilament fibers with small diameters and long lengths. The fibers were intertwined with each other, and the length–diameter ratio was large. In the ordinary corn stalk fiber, there were both fiber bundles and monofilament fibers. The diameter difference was large, the fiber ends could clearly be seen, and the length–diameter ratio was small. After the modification of corn stalk fiber by alkali solution, the fiber bundle structure showed the phenomenon of filamentation and fiber refinement. As the quantity of monofilament fibers increased, the aspect ratio increased, and the fiber's surface became increasingly rough, which, in turn, increased the area of contact between the asphalt and the fiber.

### 2.2. Mix Proportion Design

#### 2.2.1. Design of Gradation Composition for Mineral Aggregate

In order to better compare the impact of fiberless, lignin fiber, and modified corn stalk fiber on the asphalt mixture's performance, AC-13 synthetic gradation was adopted, as depicted in Figure 3.

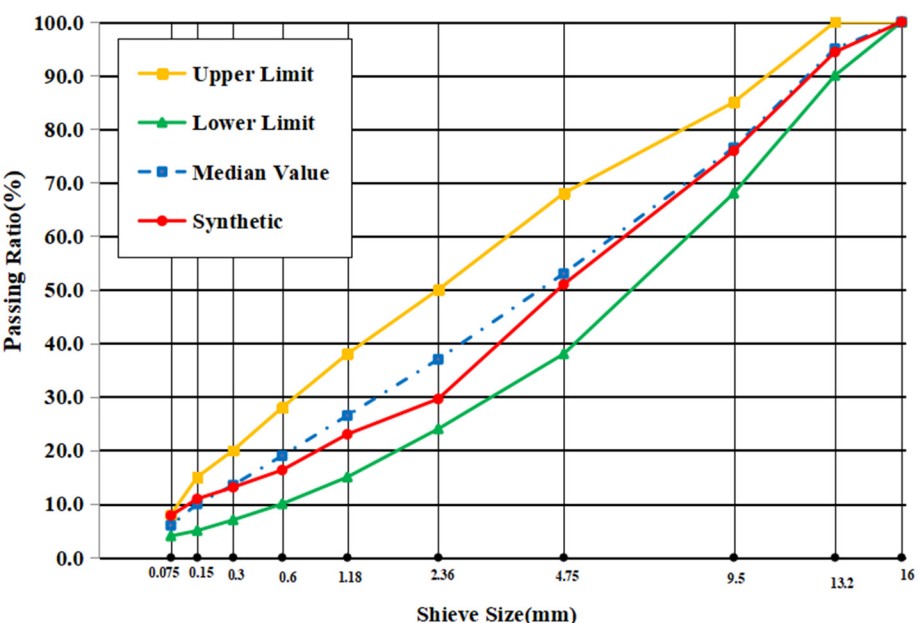

**Figure 3.** Gradation composition of AC-13 asphalt mixture.

#### 2.2.2. Fiber Asphalt Mixture Mixing Process

Fiber asphalt mixture was mixed in two different ways, the dry mix and wet mix methods [36]. The dry mix method is to mix the coarse and fine aggregate with the fiber, then add the asphalt, and, finally, add the mineral powder. The wet mixing method is to mix fiber and asphalt first to make fiber asphalt mortar and then pour it into the mixing pot to mix with aggregate. Due to the complexity of the process of preparing fiber asphalt via the wet mixing method, as well as the not timely nature of mixing and ease of the occurrence of the segregation phenomenon, other equipment was needed to prepare fiber asphalt, which led to an increase in construction costs and a decrease in efficiency. So, this paper selected the dry mixing method. First, the fibers were uniformly placed in the aggregate at 160 °C for 90 s. Then, a certain amount of asphalt was poured in and mixed for 90 s. Finally, the ore powder was poured into the continuous mixing for 90 s. When the internal temperature of the fiber asphalt mixture was 140 °C, the double-sided compaction was 75 times. Figure 4 depicts the mixing procedure.

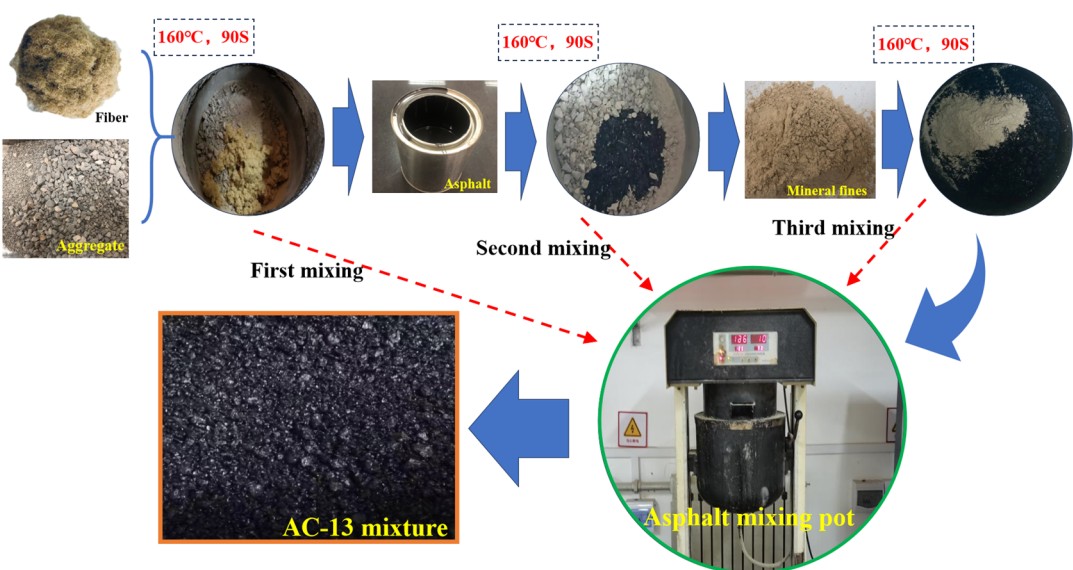

**Figure 4.** The fiber asphalt mixture mixing process.

2.2.3. Determination of Optimum Asphalt–Aggregate Ratio of Asphalt Mixture

In order to calculate the optimum asphalt–aggregate ratio of fiberless asphalt mixture, the Marshall test method was used to test the asphalt–aggregate ratios of 4.0%, 4.5%, 5.0%, 5.5%, and 6.0%. Four Marshall specimens were prepared for each group. On the basis of the Chinese Standard JTG E20-2011 [34], the gross volume relative density $\gamma f$, Marshall stability MS, and flow value FL of the specimens were measured, and the void ratio (VV), mineral aggregate void ratio (VMA), and effective asphalt saturation (VFA) were calculated. Table 3 presents the test results.

**Table 3. The** asphalt mixture's fiberless test results.

| Asphalt–Stone Ratio (%) | $\gamma f$ (g/cm$^{-3}$) | VV (%) | VMA (%) | VFA (%) | MS (KN) | FL (mm) |
|---|---|---|---|---|---|---|
| 4.0 | 2.410 | 6.1 | 14.1 | 57.0 | 11.43 | 29.6 |
| 4.5 | 2.416 | 5.2 | 14.4 | 64.0 | 11.72 | 32.4 |
| 5.0 | 2.420 | 4.3 | 14.6 | 70.5 | 11.56 | 23.7 |
| 5.5 | 2.430 | 3.2 | 14.7 | 77.9 | 10.95 | 32.0 |
| 6.0 | 2.369 | 5.0 | 16.2 | 70.8 | 11.18 | 30.4 |

According to Table 3, the trend in the relationship between the oil–rock ratio and the indicators of the Marshall test was analyzed, and the optimum oil–rock ratio $OAC_1$ was found to be 4.96%. The median $OAC_2$ of the oil–stone ratio range that meets the specification requirements was 4.89%. Therefore, the optimum asphalt–aggregate ratio of fiber-free asphalt mixture was 4.93%.

According to the calculation method of the optimum oil–stone ratios of the fiber-free asphalt mixture, modified corn stalk fiber asphalt mixture, and lignin fiber asphalt mixture with the content of 0.1%, 0.2%, 0.3% and 0.4% were tested via the Marshall test. The optimum mixture's asphalt–aggregate ratios under different fiber contents were calculated, and the examination results for each index under the asphalt–aggregate ratio were measured, as depicted in Table 4.

**Table 4.** Marshall test results of fiber asphalt mixture under optimum asphalt–aggregate ratio.

| Mixture Type | Fiber-Free | Modified Corn Stalk Fiber | | | | Lignin Fiber | | | |
|---|---|---|---|---|---|---|---|---|---|
| Fiber content (%) | 0 | 0.1 | 0.2 | 0.3 | 0.4 | 0.1 | 0.2 | 0.3 | 0.4 |
| The ideal amount of asphalt (%) | 4.93 | 5.18 | 5.25 | 5.36 | 5.52 | 5.06 | 5.17 | 5.32 | 5.41 |
| Bulk density (g/cm$^3$) | 2.433 | 2.421 | 2.450 | 2.432 | 2.441 | 2.435 | 2.437 | 2.443 | 2.399 |
| Percentage of void (%) | 5.2 | 3.5 | 3.8 | 3.8 | 2.8 | 4.7 | 4.4 | 3.4 | 4.5 |
| Gap rate of mineral (%) | 14.1 | 14.1 | 13.7 | 14.4 | 14.2 | 14.3 | 14.2 | 14.1 | 15.7 |
| Asphalt saturation (%) | 66.3 | 70.8 | 72.1 | 73.6 | 80.4 | 67.5 | 69.2 | 75.7 | 71.5 |
| Degree of stability (KN) | 12.37 | 13.28 | 14.34 | 14.99 | 11.77 | 13.56 | 14.32 | 12.98 | 12.04 |
| Flow value (0.1 mm) | 30.9 | 31.5 | 35.0 | 36.6 | 29.1 | 32.4 | 35.1 | 28.3 | 36.9 |

### 2.3. Test Scheme

Figure 5 depicts the experimental flowchart.

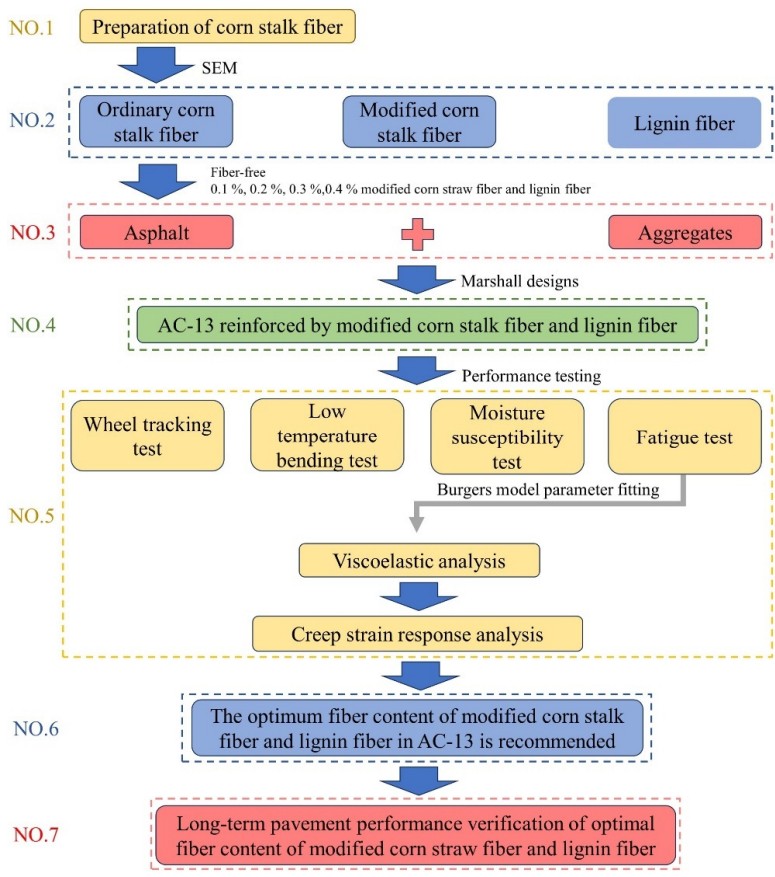

**Figure 5.** Experimental flowchart.

### 2.3.1. Wheel Tracking Test

As a kind of viscoelastic material, the asphalt mixture is affected by temperature changes in its mechanical properties. In summer, the temperature is as high as 60 °C, and asphalt pavement undergoes permanent deformation under long-term load and high temperatures [37]. In this paper, the T0719-2011 rutting test method in the Chinese Standard (JTG E20-2011) [34] is adopted to evaluate the asphalt mixtures' resistance to high temperatures. The rut specimens of the lignin fiber asphalt mixture and modified corn stalk fiber asphalt mixture with dosages of 0%, 0.1%, 0.2%, 0.3%, and 0.4% and the size of 300 mm × 300 mm × 50 mm were created, with a total of 13 groups, with 3 in each group. The rutting specimens and test molds were positioned in a constant temperature chamber at a temperature of 60 °C for 5 h. The heat-insulated rut specimen was placed on the rut test bench. The test wheel was situated in the center of the rut plate. The contact pressure

with the specimen was 0.7 MPa, and the round-trip rolling speed was 42 times/min, which lasted for 1 h or until the maximum deformation exceeded 25 mm. According to the rutting test machine, the rutting deformation $d_1$ and $d_2$ of 45 min ($t_1$) and 60 min ($t_2$) were automatically read, and the dynamic stability was calculated according to Equation (1).

$$DS = \frac{(t_2 - t_1) \times N}{d_2 - d_1} \times C_1 \times C_2 \tag{1}$$

where $DS$—The asphalt mixture's dynamic stability, times/mm; $d_1$—The deformation corresponding to time $t_1$, mm; $d_2$—The deformation corresponding to time $t_2$, mm; $C_1$—The test machine type parameters, take 1.0; $C_2$—The specimen coefficient, take 1.0.; $N$—The test wheel typically rolls at a speed of 42 times per minute.

### 2.3.2. Low-Temperature Bending Test

Based on the T0715-2011 asphalt mixture bending test method in the Chinese Standard (JTG E20-2011) [34], the impact of fiber type and dosage on the low-temperature performances of asphalt mixtures was evaluated. For rutting specimens of asphalt mixture without fiber, four dosages of lignin fiber asphalt mixture and modified corn stalk fiber asphalt mixture were made, respectively, and they were cut into prismatic beams of 250 mm × 30 mm × 35 mm. Before the test, the trabecular specimens spent four hours in a temperature-controlled enclosure at −10 °C for 4 h. Subsequently, the insulated prismatic beamlets were tested using AST equipment with a 50 mm/min loading rate. In accordance with Equation (2), the bending strength modulus was calculated.

$$S_\text{B} = \frac{L^3 \times P_B}{4 \times b \times h^3 \times d} \tag{2}$$

where $S_\text{B}$—The specimen's modulus of bending stiffness at failure, MPa; $L$—The span of specimen, mm; $P_B$—The maximum load when the specimen is destroyed, N; $b$—The width of the cross-interrupted interview, mm; $h$—The height of interrupted interview; $d$—The mid-span deflection at failure, mm.

### 2.3.3. Moisture Susceptibility Test

Based on the test methods T0709-2011 and T0729-2000 in the Chinese Standard (JTG E20-2011) [34], the impact of fiber type and dosage on the water stability of the asphalt mixtures were evaluated. Two groups of Marshall specimens with 75 times compaction were immersed in a 60 °C constant temperature water bath for 30 min and 48 h, respectively, and then placed in an automatic Marshall tester for the immersion residual stability test. For the Marshall specimens with each side compressed 50 times, the first group was subjected to a 97.3–98.7 Kpa vacuum saturated with water for 15 min and immersion in water at atmospheric pressure for 30 min, before being taken out and put into a plastic bag with 10 mL of water in a thermostat at −18 °C and frozen for 16 h, and then we removed the plastic bag and kept it warm in a thermostatic water bath at a temperature of 60 °C for 24 h. Ultimately, the splitting strengths of the two sets of Marshall specimens were evaluated after they were placed in a bath at a temperature maintained at a constant of 25 °C for two hours, and the splitting strength was tested. The test loading rate was 50 mm/min. On the basis of the freeze–thaw splitting test and immersion Marshall test results, the immersion residual stability and freeze–thaw splitting tensile strength ratio of the fiber asphalt mixture were calculated according to Equations (3) and (4).

$$MS_0 = \frac{MS_1}{MS} \times 100 \tag{3}$$

where $MS_0$—The specimen's residual stability in water, %; $MS$—The stability of the specimens after 30 min of water immersion, KN; $MS_1$—The stability of the specimens after 48 h of water immersion, KN.

$$TSR = \frac{\overline{R}_{T1}}{\overline{R}_{T2}} \times 100 \qquad (4)$$

where $TSR$—The freeze–thaw splitting test's strength ratio,%; $\overline{R}_{T1}$—The splitting tensile strength of the first group of single specimens subjected to freeze–thaw cycles, MPa; $\overline{R}_{T2}$—The splitting tensile strength of the second group of single specimens without freeze–thaw cycles.

### 2.3.4. Fatigue Test

In this paper, a fatigue test was carried out by using the AST test equipment and stress loading mode. The asphalt mixture's fatigue performance index was determined by the number of loading times at failure [38]. A total of 9 groups of lignin fiber asphalt mixture specimens and modified corn stalk fiber asphalt mixture trabecular specimens with contents of 0%, 0.1%, 0.2%, 0.3%, and 0.4% were made, with 6 in each group. The temperature used for testing was 15 °C, and the dimensions of the specimen were 250 × 30 × 35 mm. The beam specimens were loaded with a half-sine wave with a wave width of 200 ms. The first group of specimens was tested for failure load values. The second group of specimens was subjected to a three-point bending test. The loading stress value of three-point bending test was 15% of the failure load. The termination condition of this test was that the deflection of the specimen increased to 9 mm in mid-span. The loading times were calculated according to the test time when the termination condition was reached.

### 2.3.5. Long-Term Pavement Performance Test

The pavement long-term performance monitoring equipment independently developed by the group of investigators was equipped with heating devices and equipment loading units. It could simulate the asphalt mixture's resistance to rutting under the combined action of different temperatures (20–65 °C) and different driving loads. It can also assess the ability of the asphalt mixture to resist long-term deformation under single environmental conditions. The pavement long-term performance monitoring equipment is shown in Figure 6.

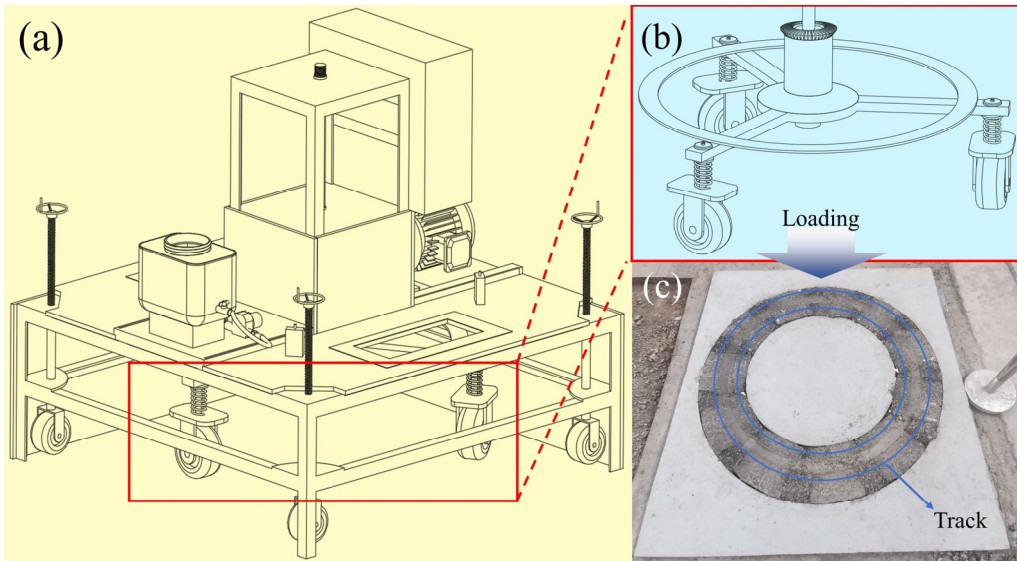

**Figure 6.** Pavement long-term performance monitoring equipment. (**a**) The actual diagram of the equipment. (**b**) Loading wheelset device. (**c**) Test track.

The long-term performance test of the asphalt pavement could evaluate the rutting resistance of an asphalt pavement under long-term high-temperature and traffic loads. The rutting specimens with the best fiber contents of different fiber types were used to

verify the resistance to permanent deformation under long-term traffic loads by using the long-term performance monitoring equipment independently developed by the research group. Three different temperatures were used for the test: 30, 40, and 50 °C. The wheel contact pressure was 0.7 MPa, and the rotation speed was 7 r/min on the paved test ring. The device recorded the rut depth data once every 3000 turns. When the number of turns of the device reached 15,000, the test stopped.

## 3. Test Results and Discussion

### 3.1. Road Performance of Fiber Asphalt Mixture

3.1.1. High-Temperature Stability

Figure 7 displays the results of a rutting test of the asphalt mixture with various fiber types and contents. Figure 7 shows that the asphalt mixture with fiber has more dynamic stability than the asphalt mixture without fiber, and all of them satisfy the specifications. This indicates that adding fiber to the asphalt mixture can improve its resistance to deformation at high temperatures. With the increase in fiber content, the dynamic stability of the modified corn stalk fiber asphalt mixture and lignin fiber asphalt mixture first increases and then decreases. The dynamic stability of the lignin fiber asphalt mixture and modified corn stalk fiber asphalt mixture achieved maximum values at 0.3% and 0.2% dosages, respectively. Among them, the high-temperature stability of modified corn stalk fiber with a 0.2% content is the best, and its dynamic stability is 19.9% and 47.0% higher than that of lignin fiber with a 0.3% content and asphalt mixture without fiber. Compared with the unmodified corn stalk fiber, the pH value and oil absorption ratio of the modified corn stalk fiber are improved, and the surface roughness of the fiber is increased, which has positive effects on free asphalt adsorption. Relatively stable structural asphalt is formed on the surface of the fiber, which improves the asphalt's viscosity and effectively reduces the fluidity of the asphalt at high temperatures. The aggregate is bonded in the mixture to form a frame structure, thereby reducing the slip between the aggregates in the mixture [39,40]. At the same time, corn stalk fiber is simpler to spread in asphalt mixture due to its shorter length compared to lignin fiber, which promotes the formation of a three-dimensional network structure. It can effectively disperse the stress to reduce the role of concentrated loads, enhancing asphalt mixtures' bearing capabilities, thus improving the high-temperature stability of asphalt mixtures.

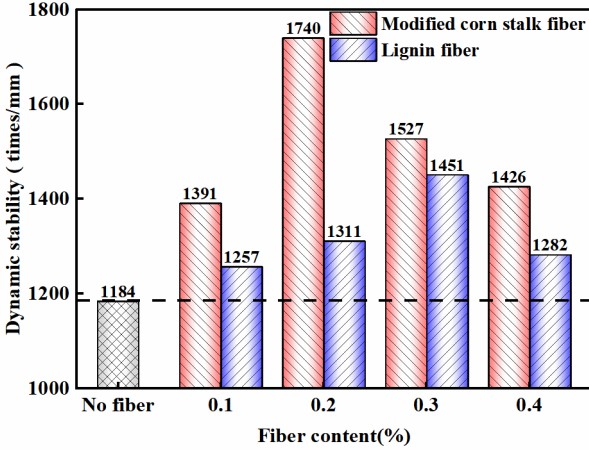

**Figure 7.** Rutting test results of asphalt mixtures with different fiber types and dosages.

3.1.2. Low-Temperature Crack Resistance

Figure 8 illustrates the asphalt mixture's low-temperature trabecular bending test results. As displayed in Figure 8, with the increase in the fiber content, the fiber asphalt mixture's bending stiffness modulus development trend is largely consistent with its dynamic stability development trend, and the bending stiffness modulus of the modified corn stalk fiber asphalt mixture and lignin fiber asphalt mixture reach the maximum value

at contents of 0.2% and 0.3%. At the same time, modified corn stalk fiber asphalt mixture with a fiber content of 0.2% has the best effect on improving the low-temperature crack resistance of the asphalt mixture. Compared with the asphalt mixture without fiber and the lignin fiber asphalt mixture with a 0.3% content, the flexural modulus increased by 28.72% and 18.28%, respectively. The modified corn stalk fibers had lower ash contents than the unmodified corn stalk fibers and lignin fibers, and the fiber breaking strength and elasticity increased, which improved the fibers' own tensile strengths. At the same time, a three-dimensional network skeleton structure [33] is formed by the fiber within the asphalt mixture, which plays a role in reinforcing and sharing stress in the asphalt mixture, slowing down the development of cracks, thus enhancing the asphalt mixture's resilience to cracking at low temperatures.

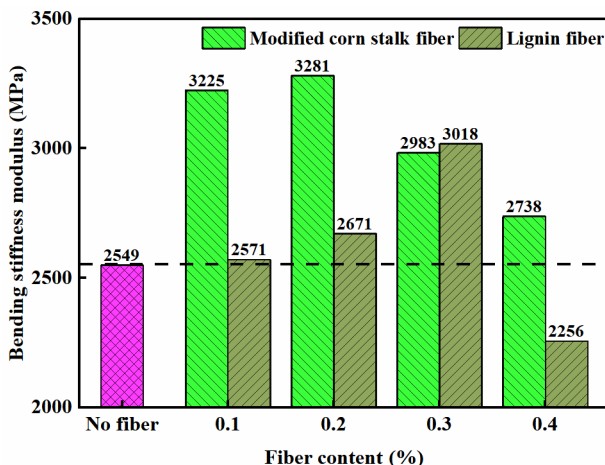

**Figure 8.** Trabecular bending test results of asphalt mixtures with different fiber types and dosages.

3.1.3. Moisture Susceptibility

The test findings presented in Figure 9 demonstrate that the addition of fibers improves the asphalt mixture's residual stability and freeze–thaw splitting ratio. Under different fiber contents, the modified corn stalk fiber asphalt mixture exhibits a superior residual stability and freeze–thaw splitting ratio in comparison to the lignin fiber asphalt mixture. Meanwhile, the freeze–thaw split strength ratios of the modified corn stalk fiber asphalt mixtures and lignin fiber asphalt asphalt mixtures both showed a tendency to rise initially before falling as the fiber contents increased. When modified corn stalk fiber asphalt mixture and lignin fiber asphalt mixture have respective contents of 0.3% and 0.2%, the freeze–thaw splitting strength ratio reaches the maximum levels, which are 89.79% and 97.52%, respectively. The residual stability of the asphalt mixture is not significantly affected by fiber incorporation. When the contents of modified corn stalk fiber and lignin fiber are 0.4% and 0.2%, respectively, the dynamic stability is the best. Compared with the modified corn stalk fiber asphalt mixture with a content of 0.2% and the lignin fiber asphalt mixture with a content of 0.3%, the residual stability is only increased by 1.2% and 1.7%. Combining their properties, the modified corn stalk fiber and lignin fiber had the best effects on water stability enhancement in asphalt mixtures at dosages of 0.2% and 0.3%, respectively. Compared to the lignin fiber asphalt mixture with a 0.3% content, the modified corn stalk fiber asphalt mixture with a 0.2% content has superior water stability. Its dynamic stability is 7.65% and 4.19% higher than those of the undoped fiber asphalt mixture and the lignin fiber asphalt mixture with a content of 0.3%, and its freeze–thaw splitting strength ratio is increased by 15% and 8.6%, respectively. The oil absorption rate and surface roughness of the modified corn stalk fiber are improved, which is beneficial to the adsorption of free asphalt by the fiber, and the proportion of structural asphalt in the asphalt mixture is increased, so that the adhesion between asphalt and aggregate is enhanced. When water enters the internal pores of asphalt and aggregate through the asphalt mixture, the aggregate and asphalt are not easily prone to spalling. At the same

time, the modified corn stalk fiber exerts its own advantages, and the reinforcement and stress sharing effect in the asphalt mixture is significant, as it can enhance the modified corn stalk fiber asphalt mixture's water stability by preventing the formation of microcracks and further reducing the amount of water entering the mixture's internal pores [41].

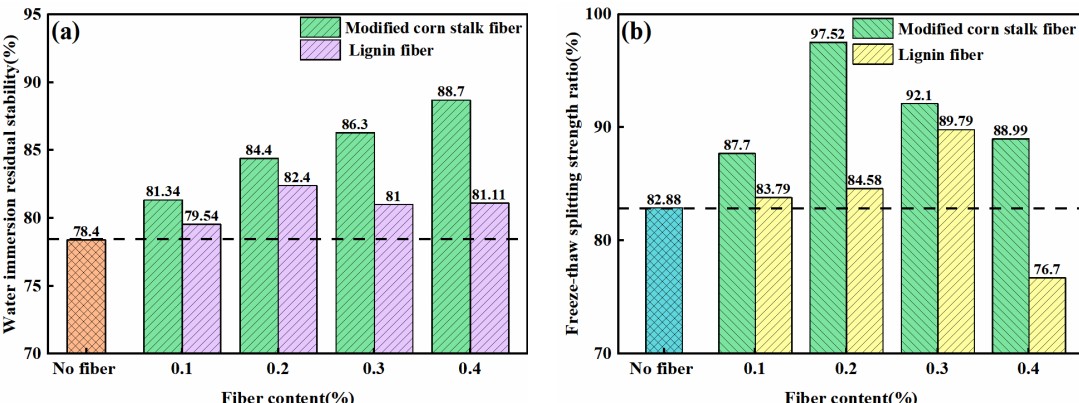

**Figure 9.** Water stability test results of asphalt mixtures with different fiber types and dosages. (**a**) Water immersion residual stability. (**b**) Freeze–thaw splitting strength ratio.

### 3.1.4. Fatigue Performance

Figure 10 shows the results of the three-point bending. According to the loading times of failure in Figure 10, it is evident that in contrast to the asphalt mixture without fiber, the loading times of the asphalt mixture with fiber are significantly improved. The incorporation of fiber can significantly enhance the anti-fatigue deformation ability of the asphalt mixture and extend the asphalt pavement's lifespan. With the increase in the fiber content, the loading times of modified corn stalk fiber and lignin fiber asphalt mixture increased first and then decreased, and when the contents were 0.2% and 0.3%, the loading times were the largest, 7425 times and 6792 times, respectively. It is evident that the modified corn stalk fiber asphalt mixture with a dosage of 0.2% has the best fatigue resistance, which is 76.7% and 9.3% higher than those of the unadulterated fiber asphalt mixture and the lignin fiber asphalt mixture with a dosage of 0.3%, respectively. Compared with lignin fiber, the tensile strength and elastic modulus of modified corn stalk fiber are higher, and the elastic component ratio of asphalt mortar is improved. At the same time, the modified corn stalk fibers are evenly distributed in the asphalt mixture, forming a fiber skeleton network structure, which is able to resist large concentrated stress and alleviate the development of fatigue cracks.

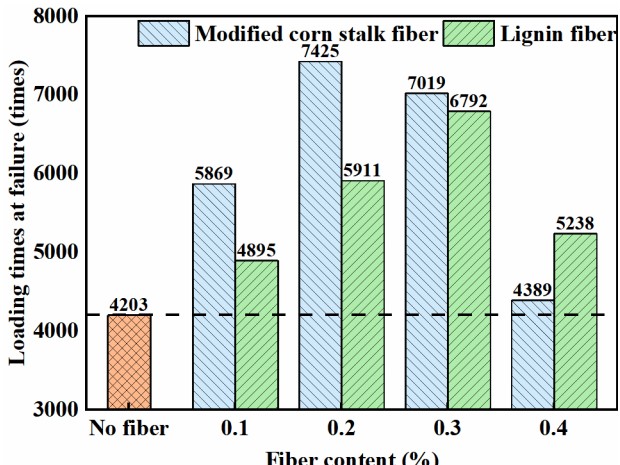

**Figure 10.** Results of tests of the fatigue performances of asphalt mixtures with different fiber types and dosages.

### 3.1.5. Viscoelastic Properties and Creep Strain Response Analysis of Fiber Asphalt Mixture

1.   Asphalt constitutive relation and the Burgers model

The viscoelastic characteristics of the asphalt mixtures are characterized by the combination of multiple basic mechanical elements such as elastic elements (springs) and viscous elements (sticky pots). The common viscoelastic models include the Kelvin, Maxwell, and Burgers models and so on [42–44]. Among them, the Burgers model is obtained by connecting the Kelvin model and the Maxwell model in series, which can well reflect the creep and relaxation characteristics of the asphalt mixture [45,46], as seen in Figure 11.

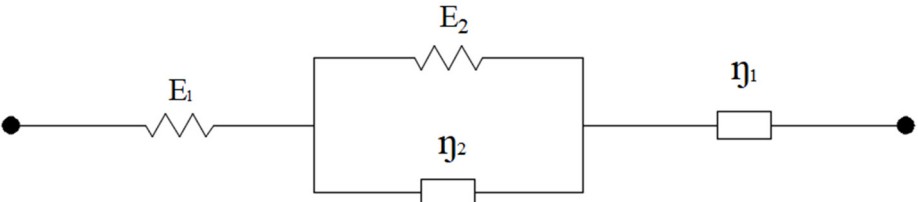

**Figure 11.** The Burgers model diagram.

The Burgers model differential constitutive equation [47] is

$$\sigma + p_1 \dot{\sigma} + p_2 \ddot{\sigma} = q_1 \dot{\varepsilon} + q_2 \ddot{\varepsilon} \tag{5}$$

where $P_1 = (\eta_1 E_1 + \eta_2 E_2 + \eta_2 E_1)/E_1 E_2$; $P_2 = \eta_1 \eta_2 / E_1 E_2$; $q_1 = \eta_1$; $q_2 = \eta_1 \eta_2 / E_2$.

When the Burgers model is subjected to a constant external stress, its constitutive relationship is derived and calculated, and the creep relation is obtained:

$$\varepsilon(t) = \sigma_0 \left[ \frac{1}{E_1} + \frac{t}{\eta_1} + \frac{1}{E_2} (1 - e^{-E_2 t / \eta_2}) \right] \tag{6}$$

where $E_1$—The Maxwell model's elastic modulus of spring; $\eta_1$—The viscosity coefficient of sticky pot in the Maxwell model; $E_2$—The Kelvin model's elastic modulus of spring; $\eta_2$—The viscosity coefficient in the Kelvin model.

2.   Viscoelastic parameter fitting and analysis

To study the effects of the fiber type and content on the viscoelasticity of the asphalt mixture, the L-M method and the general global optimization algorithm were used to fit the time and strain data obtained from the three-point bending test using the Burgers model [48], and the four viscoelastic parameters were obtained as $E_1$, $\eta_1$, $E_2$, and $\eta_2$, respectively. According to $E_2$ and $\eta_2$, the delay time $\tau = \eta_2 / E_2$ is calculated to characterize the important internal time characteristics of the viscoelastic properties of the asphalt mixture. Table 5 presents the viscoelastic parameters, and the relationship between the fiber content and viscoelastic parameters of the asphalt mixture is shown in Figure 11.

**Table 5.** Viscoelastic parameters of fiber asphalt mixture.

| Fiber Type | | No Fiber | Modified Corn Stalk Fiber | | | | Lignin Fiber | | | |
|---|---|---|---|---|---|---|---|---|---|---|
| Fiber content (%) | | 0 | 0.1 | 0.2 | 0.3 | 0.4 | 0.1 | 0.2 | 0.3 | 0.4 |
| | E1 | 8.691 | 6.434 | 3.576 | 5.471 | 6.645 | 7.435 | 6.751 | 3.610 | 7.276 |
| | η1 | 109.842 | 136.495 | 184.355 | 165.561 | 145.232 | 126.120 | 147.299 | 168.644 | 125.855 |
| Viscoelastic | E2 | 0.883 | 1.216 | 1.762 | 1.416 | 1.167 | 1.121 | 1.212 | 1.681 | 1.085 |
| parameters | η2 | 54.217 | 87.073 | 145.363 | 107.494 | 85.975 | 76.908 | 96.575 | 136.932 | 87.575 |
| | η2/E2 | 61.401 | 71.606 | 82.499 | 75.914 | 73.672 | 68.607 | 79.682 | 81.459 | 80.714 |
| | R2 | 0.995 | 0.987 | 0.985 | 0.976 | 0.972 | 0.992 | 0.984 | 0.977 | 0.971 |

In Figure 12, it is evident that as the fiber content increase, the delayed elastic parameters $E_2$ and viscous flow parameters $\eta_1$ and $\eta_2$ of modified corn stalk fiber and lignin fiber

all exhibit a growing and subsequently diminishing trend, while the instantaneous elastic parameters $E_1$ show a tendency that is initially declining and then rising.

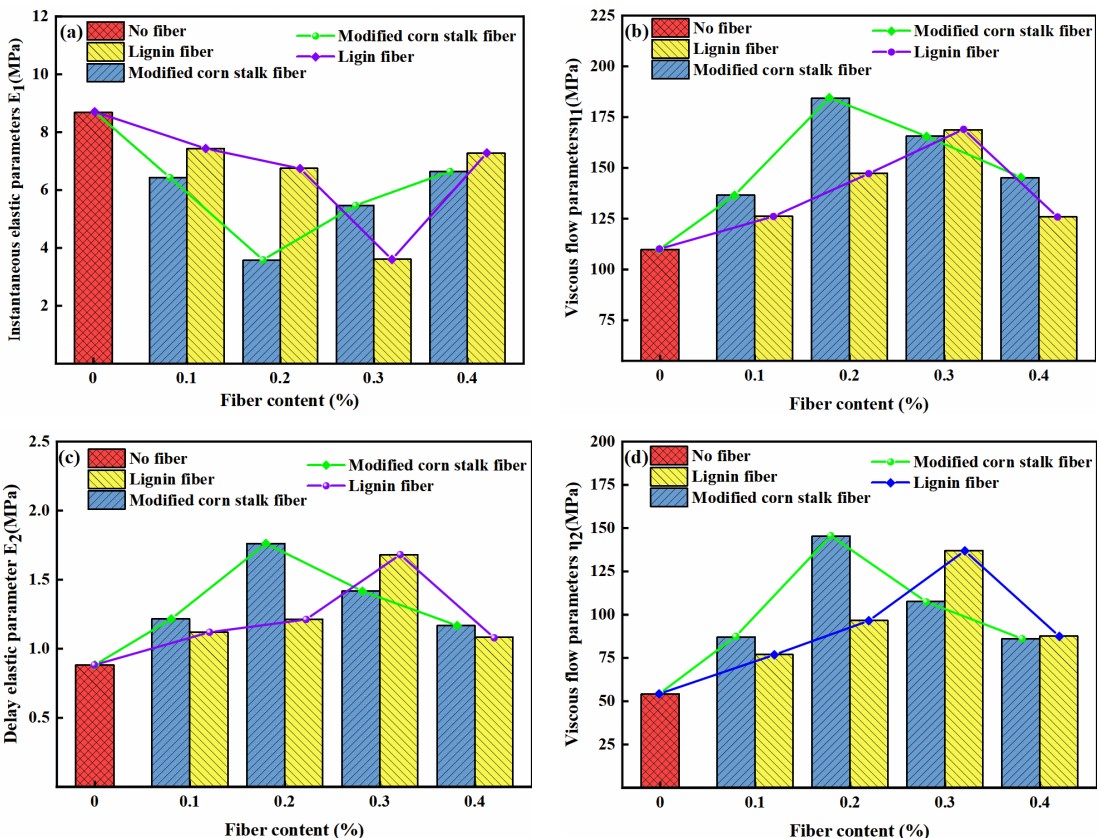

**Figure 12.** Relationship between fiber content and viscoelastic parameters of asphalt mixture. (**a**) $E_1$. (**b**) $\eta_1$. (**c**) $E_2$. (**d**) $\eta_2$.

The delay times of the asphalt mixtures with different fiber types and contents are calculated by the delay elastic parameter $E_2$ and viscous flow parameter $\eta_2$, and Figure 13 illustrates the change rule.

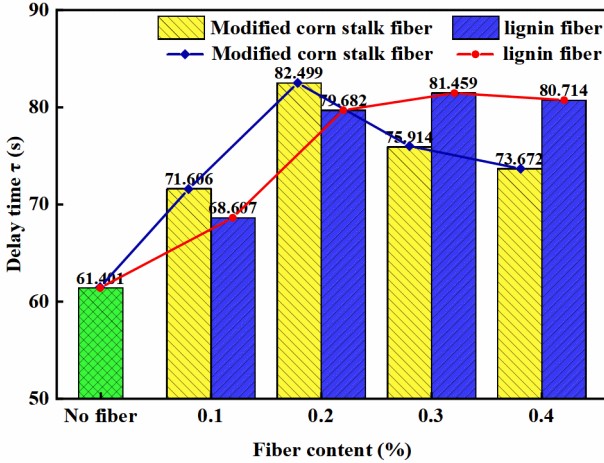

**Figure 13.** Delay time test results.

The delay times of asphalt mixtures with different fiber types and contents increase first and then decrease with the increase in fiber content. The delay times of the lignin fiber asphalt mixture and modified corn stalk fiber asphalt mixture reached the peak value when

the contents were 0.3% and 0.2%, respectively, and the delay time of the modified corn stalk fiber asphalt mixture was the longest. The results showed that adding 0.2% modified corn stalk fiber to asphalt mixture could reduce the accumulated viscoelastic strain and improve the resistance to rutting.

3.  Creep strain response analysis

To study the change law of creep behavior of the asphalt mixtures with different fiber types and contents under dynamic load, the fitting four parameters are substituted into the creep equation, and the dynamic load $\sigma_0 \sin t$ is applied via MATLAB 2022 software (where $\sigma_0$ is 0.7 MPa). The obtained strain response curve is shown in Figure 14.

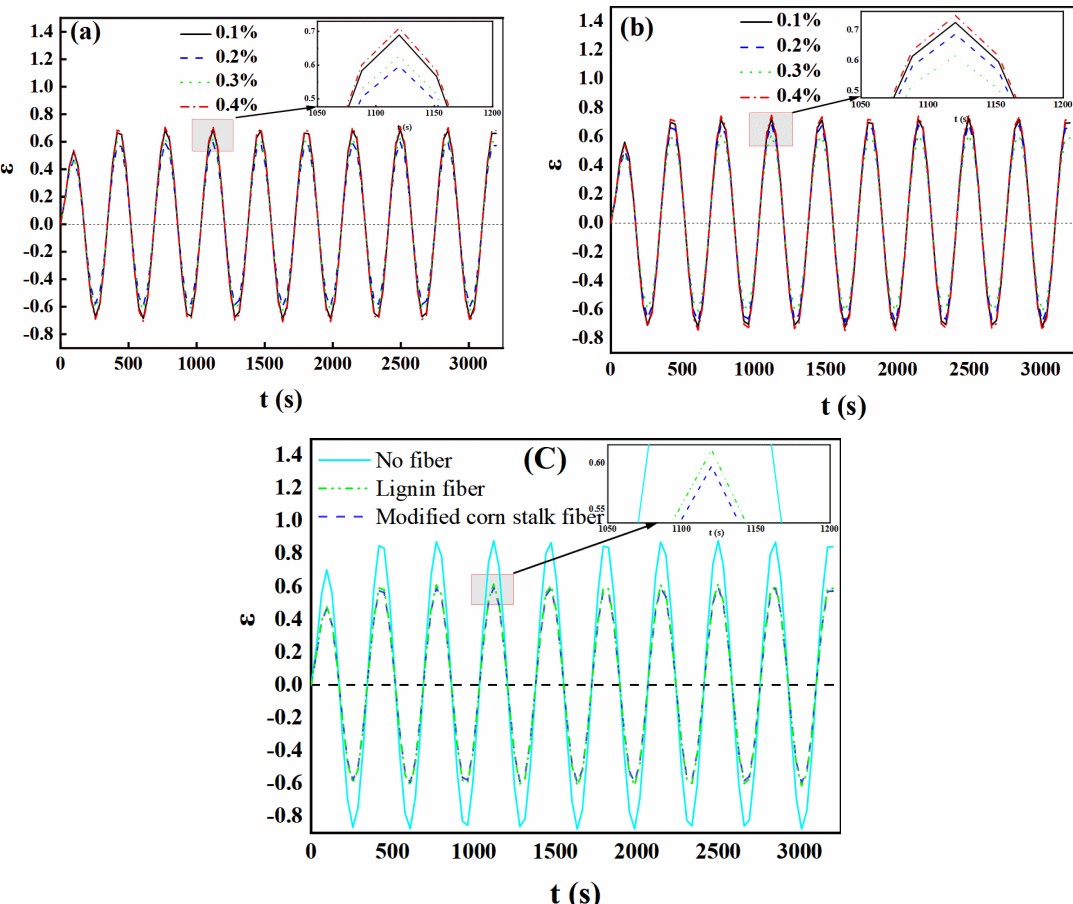

**Figure 14.** Strain response curve of fiber asphalt mixture. (**a**) Modified stalk fiber asphalt mixture. (**b**) Lignin asphalt mixture. (**c**) Three kinds of fiber asphalt mixture.

Figure 14a,b illustrate that the strain peaks of the two fiber asphalt mixtures under dynamic load decrease first and then increase as the content of fiber increases. The asphalt mixtures with modified corn stalk fiber and lignin fiber have the smallest strain peak when the contents are 0.2% and 0.3%, respectively, indicating that the asphalt mixtures with modified corn stalk fiber and lignin fiber have the smallest total strain and the best anti-deformation performance. Figure 14c is the strain response curve of asphalt mixture with different fiber types and the optimum fiber content. Compared with the asphalt mixture containing 0.3% lignin fiber, it is found that the modified corn stalk fiber asphalt mixture with a 0.2% content has a minimum strain peak, indicating that the modified corn stalk fiber with a 0.2% content has a more favorable impact on improving the creep performance of asphalt mixture than the lignin fiber asphalt mixture with a 0.3% content.

### 3.2. Long-Term Performance of Fiber Asphalt Mixture Pavement

The rutting depth variation curves of asphalt mixtures with different fiber types at optimum dosages for different loading times are shown in Figure 15.

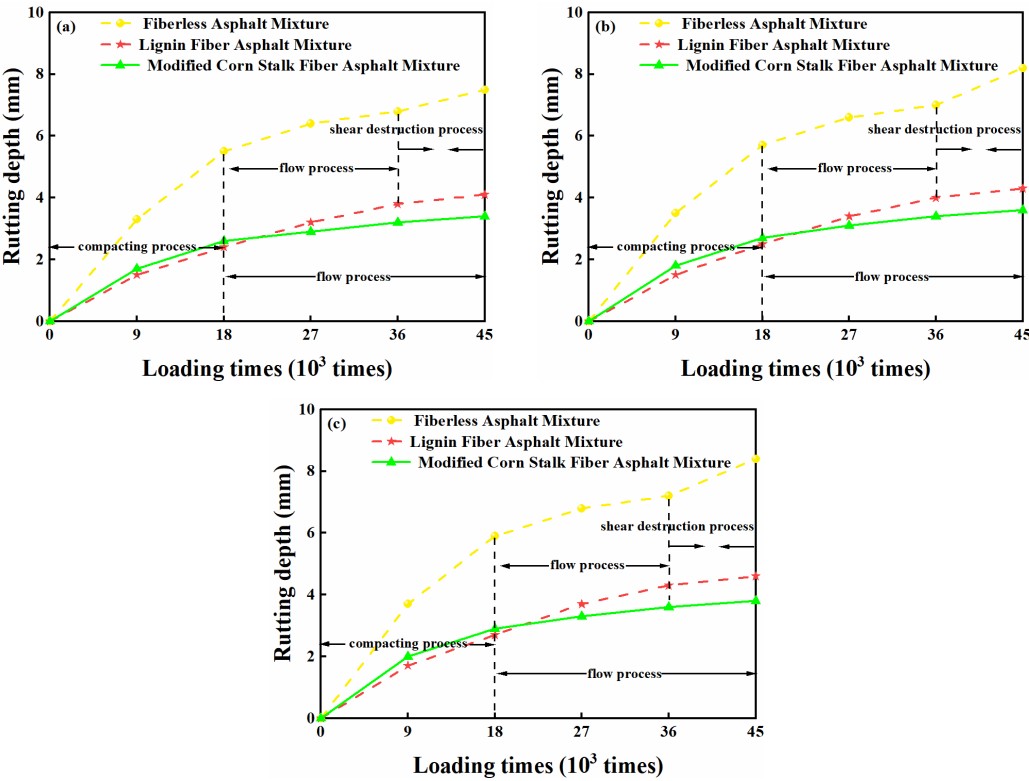

**Figure 15.** Rutting depth change curve. (**a**) 40 °C. (**b**) 50 °C. (**c**) 60 °C.

Figure 15 shows the rutting depth variation curves of unadulterated fibers, modified corn stalk fiber asphalt mixture with a 0.2% dosage, and lignin fiber asphalt mixture with a 0.3% dosage at different temperatures. Under the same test parameters such as test temperature, wheel speed, and wheel load, the undoped fiber asphalt mixture undergoes three processes of compaction, flow, and shear failure, while the lignin fiber asphalt mixture and the modified corn stalk fiber asphalt mixture only undergo two processes of compaction and flow, proving that the asphalt mixture's resistance to rutting at high temperatures may be considerably increased by adding fiber. Moreover, the growth trend of the rutting depths of the two kinds of fiber asphalt mixtures is roughly the same at different loading times. When the loading times exceed 27,000 times, the rutting depth of modified corn stalk fiber asphalt mixture with a 0.2% content is less than that of lignin fiber asphalt mixture with a 0.3% content, indicating that the long-term high-temperature stability of modified corn stalk fiber is greater than that of lignin fiber, and it has more significant effects on adsorption, asphalt stabilization, and reinforcement, which makes the long-term performance of the modified corn stalk fiber asphalt mixture optimal.

### 4. Conclusions

(1) The porosity, mineral aggregate clearance, asphalt saturation, stability, and flow value of the asphalt mixture under the optimum asphalt–aggregate ratio meet the requirements of the specification. In comparison to fiber-free asphalt mixture, the fiber asphalt mixture has a higher ideal asphalt–aggregate ratio, and with the increase in fiber content, the asphalt–aggregate ratio increases gradually.

(2) Road performance tests on fiber asphalt mixtures reveal that the optimal fiber blending percentages of lignin fiber asphalt mixtures and modified corn stalk fiber asphalt mixtures were 0.3% and 0.2%, respectively. Among them, the modified corn stalk fiber with

a dosage of 0.2% has the best road performance. Its dynamic stability, bending stiffness modulus, immersion residual stability, freeze–thaw splitting strength ratio, and loading times at failure are 19.9%, 18.28%, 4.19%, 8.6%, and 9.15% higher than those of lignin fiber asphalt mixture with a dosage of 0.3%, respectively, as well as 47.0%, 28.72%, 7.65%, 15%, and 75.81% higher than those of undoped fiber asphalt mixture, respectively.

(3) Through the analysis of the Burgers model parameters, viscoelastic index, and creep strain response, it is found that the delayed elastic parameter $E_2$, viscous flow parameter $\eta_1$, $\eta_2$, delay time $\tau$, and creep strain of 0.3% lignin fiber asphalt mixture and 0.2% modified corn stalk fiber asphalt mixture have the maximum value, and the instantaneous elastic parameter $E_1$ has the minimum value. The viscoelastic properties and creep behavior of modified corn stalk fiber with a 0.2% content are better than those of lignin fiber asphalt mixture.

(4) The long-term performance of the fiber asphalt mixture is monitored, and it is found that the modified corn stalk fiber asphalt mixture with a 0.2% content has the best long-term high-temperature stability.

**Author Contributions:** K.W. and P.H. were responsible for the conception and review of the manuscript; L.Q. was in charge of writing the paper; Q.W. and H.X. were in charge of testing and data analysis; L.T. and X.Z. were in charge of manuscript review. All authors have read and agreed to the published version of the manuscript.

**Funding:** This work was supported by Science and Technology Plan of the Shandong Provincial Department of Transportation (2022B109).

**Institutional Review Board Statement:** Not applicable.

**Informed Consent Statement:** Not applicable.

**Data Availability Statement:** The article contains all the information needed to support this study's conclusions.

**Acknowledgments:** The authors express their gratitude to the School of Transportation and Civil Engineering at Shandong Jiaotong University for supplying the experimental apparatus and materials utilized in this investigation.

**Conflicts of Interest:** Authors Liang Tang and Xiaofei Zhang were employed by the company Shandong Road and Bridge Group Co., Ltd. The remaining authors declare that the research was conducted in the absence of any commercial or financial relationships that could be construed as a potential conflict of interest.

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
