# Peer review of "Investigation on the Performance of Modified Corn Stalk Fiber AC-13 Asphalt Mixture"

_coatings, doi:10.3390/coatings14040436_

Round 1

Reviewer 1 Report

Comments and Suggestions for Authors

It is estimated that this study aims to comparatively examine the effects of fiber-free and modified corn stalk fiber and lignin fiber addition on the performance of AC-13 asphalt mixture. Because the main purpose of the study is not clearly stated. It presents the results of a comprehensive, well-organized empirical study with very convincing arguments. As stated in the manuscript, there are many studies on the use of modified corn stalk fiber with the same modification method, which is considered a serious weakness in terms of the originality of the study.

However, it can be said that the selected asphalt mixture and the meaningful and frequent tests focused on determining the performance of the asphalt mixture, not directly on the asphalt mortar, can compensate for this novelty weakness to a certain extent.

My questions, comments, and suggestions about the entire manuscript are highlighted in the text and provided as pop-up notes.

Comments on the Quality of English Language

It is very good.  

Reviewer 2 Report

Comments and Suggestions for Authors

The paper deals with the use of modified corn stalk in asphalt concrete. The investigated topic is really interesting specifically in this time where the use of secondary materials is encouraging but in the Reviewer’s opinion there some  drawbacks that require a clarification. First of all, why the Marshall compaction was selected since it is well known that is totally not a simulative method of  compaction? Moreover, what was the protocol to obtain the slabs then  subjected to the rutting test? The Results section, in some parts, appears to be lacking of adequate explanations of the obtain results. Reference are quite comprehensive and recent even though not all of them seem to be perfectly appropriate. Anyway, the Reviewer considers the experimental investigation described to be exhaustive and well structured. In order to improve the obtained data, the use of the gyratory compaction could be  useful. Moreover, since the use of fiber is known to have positive effects on cracking propagation, the fracture mechanics approach and SCB test are strongly encouraged.

Comments on the Quality of English Language

In some parts of the text a language review could be necessary to avoid any difficulty in reading and understanding the text. The language review process should aim to fix grammatical mistakes and remove odd words and typo (especially wrong capital and lowercase letter).
